# WINOGRAD STRUCTURED PRUNING

## ABSTRACT

Both Winograd convolution (*WC*) and pruning can significantly reduce the computation of Convolutional Neural Network (CNN), but applying them simultaneously is challenging. For example, applying fine-grained pruning to *WC* eliminates the computational advantages from sparsity due to Winograd transformation. Moreover, integrating *WC* with filter pruning can lead to a reduction in network accuracy due to the use of large pruning unit size. To address previous challenges, this paper proposes Adaptive Balanced Winograd Structured Pruning (ABWSP), a method specifically designed to prune weights in *WC* networks executed on GPUs, which are widely used as computing devices for CNNs. ABWSP takes into account three crucial factors: pruning unit size, workload balance, and layer importance. First, ABWSP efficiently utilizes the computing units on GPUs by pruning grouped weights simultaneously. Considering the computational characteristics of *WC* on GPU, the group size can be minimized while maintaining a regular data pattern (i.e., WSP). Secondly, the General Matrix Multiplications (GEMMs) of a layer are executed concurrently on GPU, and the execution time of the layer is determined by the longest GEMM operation. To minimize the execution cycle for *WC*, ABWSP maintains an equal pruning ratio between matrices of *WC* (i.e., BWSP). Lastly, applying BWSP to all layers results in a loss of accuracy. Since the importance varies for each *WC* layer, the accuracy loss and performance benefit due to BWSP are different. To maintain accuracy with high speedup, ABWSP comprehensively evaluates both accuracy and speedup to determine the appropriate application of BWSP or WSP for each layer, automatically. By considering these factors, ABWSP optimizes the pruning process by effectively utilizing GPU computing units, minimizing execution cycles of each layer, and ensuring a balance between accuracy and speedup.

## 1 INTRODUCTION

Convolutional Neural Networks (CNNs) have emerged as a powerful tool for computer vision tasks (Krizhevsky et al., 2012; Simonyan & Zisserman, 2015). However, their high computational demands have hindered their practical application in mobile scenarios (Niu et al., 2020). To overcome this challenge, researchers have developed optimization techniques such as Winograd convolution (*WC*) (Winograd, 1980; Lavin & Gray, 2016) and pruning (Han et al., 2015), which have been widely adopted by industry leaders such as Intel, AMD, and NVIDIA. These techniques have resulted in a more than two-fold increase in the inference speed of CNNs (Yan et al., 2020; Zhu et al., 2019), prompting significant interest from the research community. This attention has subsequently fueled ongoing advancements and innovation for diverse applications (Xue et al., 2022; Andri et al., 2022).

Despite the high effectiveness of *WC* and pruning techniques in improving the performance of CNNs, using both techniques together have several challenges. Specifically, pruning is developed without considering *WC*, resulting in the removal of sparsity in fine-grained pruning (e.g. weight pruning (Han et al., 2015) and, balanced pruning (Yao et al., 2019; Mishra et al., 2021)) during the transformation process of *WC* (Liu et al., 2018). However, this does not imply that all pruning techniques are incompatible with *WC*. For instance, middle-grained pruning (e.g. tile pruning (Lin et al., 2022), block sparse (Narang et al., 2017)) and FP (Li et al., 2016; He et al., 2018) maintain such sparsity even if it is used with *WC* due to the use of larger pruning unit sizes compared to transformation unit sizes. Despite its potential benefits, the application of *WC* to middle-grained

pruning presents a challenge. This is because *WC* converts structural pruning shapes into irregular pruning shapes that require specialized hardware for effective computations, thereby limiting the speed gains achievable on commonly used computing devices such as GPUs (Yu et al., 2017; Kim et al., 2017) (see Appendix for details). Moreover, FP is a technique that does not consider the GPU calculation method of *WC*, resulting in a significant pruning unit size, which in turn leads to a loss of representation power (Mao et al., 2017) (see section 4.1.(a)).

Prior work (Liu et al., 2018) proposes a novel approach, Winograd Weight Pruning (WWP), to address the challenge of sparsity removal in *WC* resulting from the transformation process. WWP proposes a shift in the pruning timing from before to after the transformation process, effectively mitigating the issue of sparsity removal. Although WWP has demonstrated promising results in achieving high pruning ratios, its element-wise computations generate irregular data access patterns on GPUs, similar to the limitations observed with middle-grained pruning, as a result, speed gains may be restricted (Yu et al., 2017) (see section 4.1.(b)). As such, improving the performance of WWP on GPUs remains a significant challenge.

This paper introduces a technique called Adaptive Balanced Winograd Structured Pruning (ABWSP) for GPUs. ABWSP applies pruning to *WC* networks while considering three crucial factors: **the optimal pruning unit size**, **workload balance between matrices**, and **layer importance**.

**The optimal pruning unit size**: The initial step (WSP) of ABWSP involves determining the pruning unit size, which relies on the size of GPUs' execution unit. In the transformation of standard convolution to *WC*, all convolution computations are converted into element-wise multiplications. To facilitate efficient computation on GPUs, these element-wise multiplications are further transformed into multiple large-sized general matrix multiplications (GEMMs), referred to as EC2B in this paper. These transformed matrix multiplications are then scheduled on multiple streaming multiprocessors (SMs) within the GPUs. The matrix elements are grouped and executed in lock-step, following the single instruction multiple threads (SIMTs) execution paradigm. To make optimal use of these computation units, the pruning unit size is determined based on the warp and wavefront size specific to the proposed pruning.

**Workload balance between matrices**: Next, ABWSP implements a uniform pruning ratio (BWSP) for the multiple large-sized matrices. As these matrices are executed simultaneously on SMs, maintaining a comparable execution time for all GEMMs becomes crucial. In cases where the execution times differ, SMs that complete their GEMMs earlier are forced to wait for an SMs performing the longest GEMMs (Park et al., 2023). This waiting time leads to delays in executing subsequent layers. Thus, equalizing the execution time among the GEMMs helps to prevent such delays and optimize overall performance.

**Layer importance**: Finally, when BWSP is applied to all layers, there are situations where the decrease in accuracy outweighs the performance improvements. The performance gains due to BWSP vary from layer to layer. Therefore, in some layers, substituting WSP instead of BWSP does not significantly reduce performance benefits. To address this, we propose an algorithm that automatically selects WSP or BWSP on each layer using the Adaptive BN-based evaluation technique (Li et al., 2020). Our proposed algorithm simultaneously considers speedup and accuracy drop due to balance. This strategy aims to minimize the loss of representation power in networks while still achieving a significant speedup coming from both the *WC* and pruning techniques.

## 2 PRELIMINARY: WINOGRAD CONVOLUTION

**Prerequisites** The $n$ filters of $l^{th}$ convolution layer is as $\mathcal{F}^{(l)} \in \mathbb{R}^{n \times c \times k_h \times k_w}$, where $n$, $c$, $k_h$, and $k_w$ are the number of filters, the number of channels, height, and width of the filters, respectively. The $H$ and $W$ denotes the height and width of the input feature map.

### 2.1 PROCESS OF WINOGRAD CONVOLUTION

As shown in Figure 1(a) and Equation 1, there are five steps in Winograd convolution (Lavin & Gray, 2016): ① Input Transformation (ITrans), ② Filter Transformation (FTrans), ③ Element-Wise Matrix Multiplcation (EWMM), ④ Channel-wise Summation (CS), and ⑤ Output Transformation (OTrans). The $p \times p$ input patch ($d$), which is the basic block of *WC*, is extracted with stride of ($p$ -

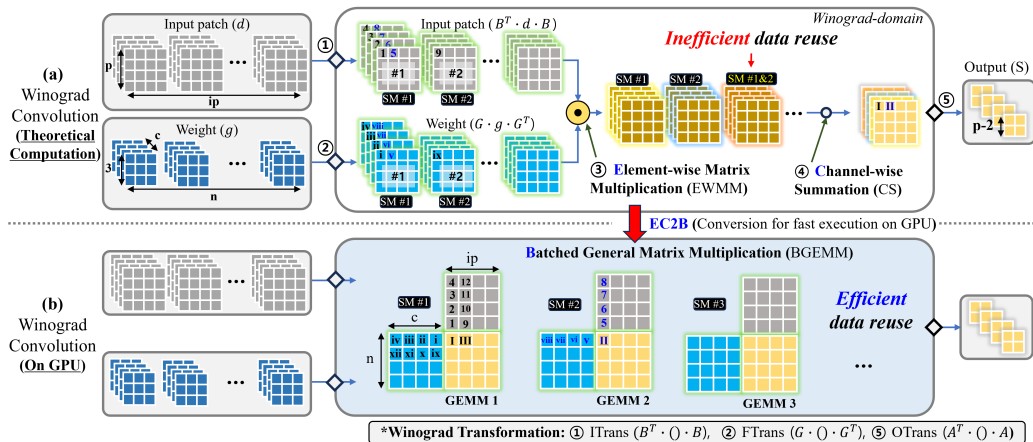

Figure 1: Overview of Winograd convolution: Conversion from theoretical computation to GPU acceleration for inference time reduction.

2) from input feature map. The input feature map has $ip$ number of input patches. The $p \times p$ input patch is convolved with a $3 \times 3$ filter ($g$) to produce an $(p\text{-}2) \times (p\text{-}2)$ output patch ($S$) (Liu et al., 2018).

$$S = A^T[(GgG^T) \odot (B^T dB)]A \tag{1}$$

ITrans and FTrans transform spatial-domain input patch ($d$) and filter ($g$) to Winograd-domain input patch ($B^T dB$) and filter ($GgG^T$) using predefined matrices $B$ and $G$, respectively (Lavin & Gray, 2016). After ITrans and FTrans, there is EWMM ($\odot$) between $B^T dB$ and $GgG^T$. Channel-wise summation reduces the channel of EWMM output. OTrans transforms Winograd-domain output patch to spatial-domain output patch $S$ by using predefined matrix $A$. The output patches are assembled into an output feature map. The matrices $B$, $G$ and $A$ used in ITrans, FTrans, and OTrans are made from Winograd's minimal algorithm (Winograd, 1980).

## 2.2 CONVERSION FOR WINOGRAD CONVOLUTION

**Inefficient process of Winograd convolution**    According to ARM (Iodice, 2018; Harris, 2014), EWMM accounts for most of the inference time of *WC*, so EWMM is a target for acceleration. However, since the EWMM results in an inefficient operation on the GPU, the parallel execution of the *WC* is not sufficient. GPU consists of dozens of independent functional modules called Streaming Multiprocessors (SMs) for parallel processing, and during EWMM on GPUs, 3D input patch ($B^T dB$) and a 3D filter ($GgG^T$) are loaded and processed in a single SM. As illustrated in Figure 1(a), to calculate filter $GgG^T$ #1 and input $B^T dB$ #1, element **i** of filter $GgG^T$ #1 is calculated only once with element **1** of input patch $B^T dB$ #1 and is not used again in the SM #1. Element **i** of filter $GgG^T$ #1 is not reused within the SM #1, so the required data (element **ix** of input patch $B^T dB$ #2) is obtained from another SM #2. However, overhead occurs when the SMs exchange data because the memory of each SM is independent.

**Efficient Winograd convolution on parallel processors**    In order to solve the low computational intensity and frequent memory access overhead of EWMM, NVIDIA (Jia et al., 2020; Yan et al., 2020) and ARM (Iodice, 2018) convert and process *WC*'s EWMM and CS to BGEMM (Batched General Matrix Multiplication) operations. BGEMM maintains a structured form to maximize data reuse on the GPU. We call this covert process as EC2B (conversion from EWMM and CS to BGEMM). As shown in Figure 1(b), BGEMM consists of a total of $p^2$ GEMM operations with the same shape. A GEMM of BGEMM is a matrix product operation between the weight matrix of $\mathbb{R}^{n \times c}$ shape and the input matrix of $\mathbb{R}^{c \times ip}$ shape. For parallelism, one GEMM is assigned to each SM. In $GEMM1$, the weight vector (**i**∼**iv**) and the input vector (**1**∼**4**) generate output element (**I**) by performing MAC operations at once. The BGEMM rearranges the input vectors (**9**∼**12**) just next the weight vectors (**1**∼**4**) in the same $GEMM1$ to increase the reuse of the weight vectors (**i**∼**iv**). EC2B can be reused as much as possible within one SM.

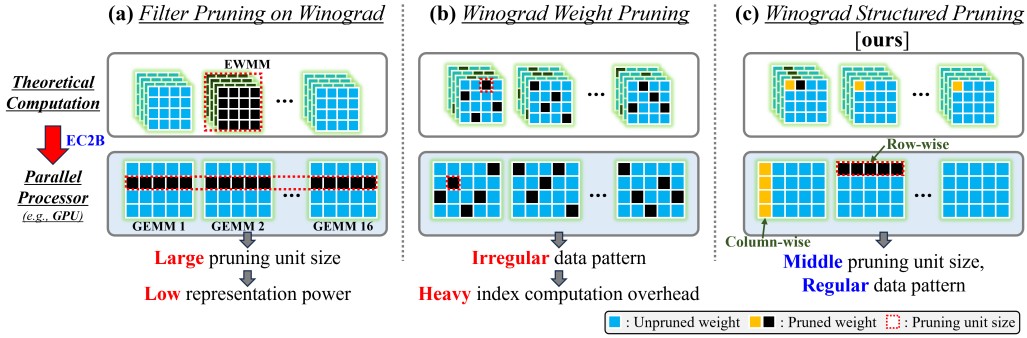

Figure 2: Previous pruning methods for Winograd convolution.

## 3 RELATED WORK

Winograd convolution (*WC*) and pruning are challenging to be compatible, but the following two previous pruning methods are compatible: FP and WWP.

**Filter Pruning (FP) on Winograd-domain**  In standard convolution, structured pruning (Li et al., 2016; Yu et al., 2018; He et al., 2017; Tanaka et al., 2020) removes rows (filter pruning) or columns (group pruning) in conversed weight matrices using im2col (Chetlur et al., 2014). According to the spatial-Winograd pruning (Yu et al., 2019), the pruning unit size is the size of the filter, which is the entire 3D filter, so the sparsity of FP is not removed by FTrans progressing in 2D units in *WC* (see Figure 2(a)). Spatial-Winograd pruning (Yu et al., 2019) reduces retraining time by pruning with FP and WWP sequentially. Since the pruned model of FP has a structural form in both standard convolution and *WC*, FP has no index computation overhead (Wen et al., 2016).

**Winograd Weight Pruning (WWP)**  WWP removes redundant Winograd-domain weight by element to reduce the parameters of Winograd Convolutional Neural Networks (see Figure 2(b)). Sparse Winograd CNN (Liu et al., 2017) proposed an approach from the perspective of *pruning* and re-training in the Winograd-domain for the first time. Winograd CNN Native Pruning (Li et al., 2017) presents the $90.0\%$ sparsity for AlexNet's Winograd parameters with less than $0.1\%$ accuracy loss in the large dataset. Winograd-ReLU CNN Pruning (Liu et al., 2018) also exploits dynamic sparsity due to activation function as well as sparsity due to weight pruning in Winograd-domain. Unfortunately, according to the yu (Yu et al., 2019), Winograd-domain dynamic sparsity of Winograd-ReLU CNN pruning (Liu et al., 2018) requires additional changes in the network structure. All the WWP methods are element-level *pruning*, which creates an irregular data pattern in Winograd-domain. The irregular data pattern requires significant index computation when accessing the pruned weights. Multiple indices cause a low throughput of the GPU owing to the memory divergence (Kloosterman et al., 2015), which requires a specialized hardware kernel or a dedicated accelerator for WWP. (Lu & Liang, 2018; Wang et al., 2019; Wu et al., 2021; Yang et al., 2021) for practical use. Therefore, we apply a hardware efficient sparsity scheme in *WC* to reduce the overhead of index computation.

## 4 PROPOSAL: ADAPTIVE BALANCED WINOGRAD STRUCTURED PRUNING

### 4.1 WINOGRAD STRUCTURED PRUNING (WSP)

**(a) Motivation 1: Large pruning unit size problem of FP**  As shown in Figure 2(a) and Equation 2, in FP, the matrix multiplication has remained in the structured form, which minimizes the index computation overhead (Wen et al., 2016), even after the EC2B conversion is applied.

$$S = A^T[(G(FilterPrune(g))G^T) \odot (B^T dB)]A \tag{2}$$

Such a regular structured form can be incredibly beneficial when general-purpose computing devices such as GPUs are used for inference. One additional benefit of the structured form is that this can be

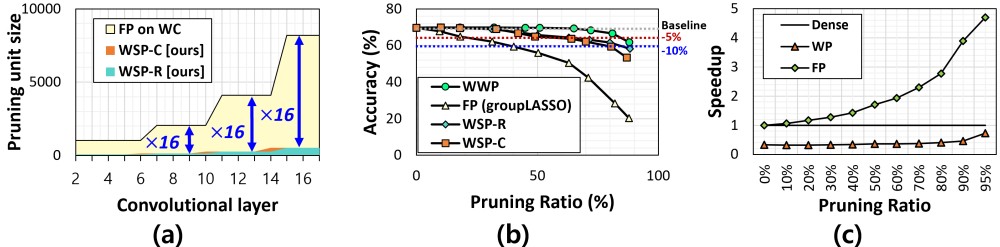

Figure 3: (a) Comparison of pruning unit size of pruned model on ResNet-18. (b) Comparison of accuracy and $PR$. Top-1 validation and $PR$ for four model on a variation of ResNet-18 on ImageNet. The fine-tuning epoch is one. (c) Comparison of speedup of GEMM by *pruning*: weight pruning (WP) and filter pruning (FP). The size of GEMM is convolution layer 6∼10 of VGG-16.

easily calculated using the GEMM of existing GPU library (Kerr et al., 2017; Nvidia, 2008), well-optimized. However, the side-effect of FP has representation loss when the models are aggressively pruned. As shown in Figure 2(a), the pruning unit size of FP on the Winograd-domain is $p^2 \times \mathbb{R}^{1 \times c}$, which tends to be too large, so it removes the important weights with non-important weights together. For example, on ResNet-18, the pruning unit sizes of FP on Winograd-domain are either $1,024$, $2,048$, $4,096$, and $8,192$ parameters depending on the layers as shown in Figure 3(a)). In Figure 3(b), when evaluating FPs with $PR$s of $20\%$ and $40\%$ in ResNet-18, the Top-1 accuracy is reduced by $5\%$ and $10\%$, respectively. Even though FP keeps the matrices in a structured form to leverage the full computing capabilities of existing general-purpose computing devices, FP shows significant accuracy drops in the high $PR$ even with retaining.

**(b) Motivation 2: Irregular data pattern problem of WWP**    As shown in Figure 2(b) and Equation 3, the WWP pruned model shows irregular data access patterns before and after applying EC2B conversion since WWP is an element-wise pruning method ($\mathbb{R}$).

$$S = A^T[WeightPrune(GgG^T) \odot (B^T dB)]A \tag{3}$$

The irregular data pattern is not adequate for parallel processors especially on GPUs (Greathouse & Daga, 2014). The irregular data pattern is calculated by converting weights into a sparse data format (e.g., CSR) to not calculate the zero values in hardware. Among the various sparse data formats, the representative CSR data format (Bell & Garland, 2009) consists of two index arrays and one non-zero value array. When calculating sparse general matrix multiplication (Sparse GEMM) using CSR, three arrays are used to obtain input data, one array is for the non-zero element row indexes, another array is for the non-zero element column indexes, and the last array is for the non-zero data variables. Using the non-zero element indexes, in Sparse GEMM, only non-zero data variables are actually computed while the operations of zero variables are simply skipped; therefore, the overall amount of computations for the Sparse GEMM is less than that of dense general matrix multiplication (GEMM). However, as shown in Figure 3(c), the inference time of pruned model (relying on the Sparse GEMM) is not faster than the inference time of the unpruned model (relying on the dense GEMM) due to the considerable amount of index computation (Hill et al., 2017; Yu et al., 2017). In order to obtain the computational benefits coming from WWP pruned models, designing specialized hardware such as NPU (Lu & Liang, 2018) is essential.

**(c) Method**    We propose Winograd Structured Pruning (WSP) that satisfies both regularity and pruning unit size as small as possible for efficient processing on GPUs. We apply structured pruning in Winograd-domain, as shown in Figure 2(c) and Equation 4. In order to preserve the sparsity by FTtrans, the WSP applies *pruning* after FTtrans. As described in Section 2.2, *WC* is transformed into BGEMM for efficient processing in GPUs (consists of 16 ($p^2$) large-sized sub-GEMMs.). To achieve the regular weight structure, our proposed WSP is removed vector by vector from each sub-GEMM weights (see Figure 4(a)). Therefore, the pruning unit size ($\mathbb{R}^{1 \times c}$) of the WSP is 16 ($p^2$) times smaller than that ($p^2 \times \mathbb{R}^{1 \times c}$) of the FP. The WSP calculates the threshold ($th$) by using the target pruning ratio ($PR$). The WSP obtains the representative value of the vector using the L2 norm method (Wen et al., 2016), and if the absolute value of the representative value is less than $th$, it is set to 0 to remove the weight vector. WSP can be divided into two approaches, row-wise WSP (WSP-R)

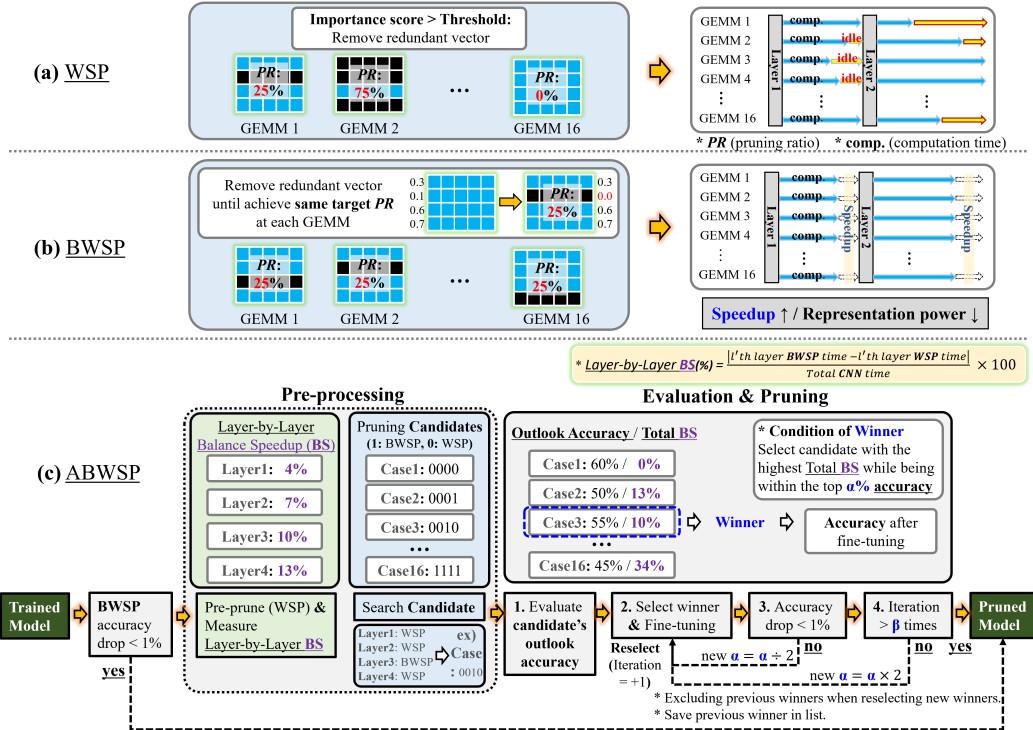

Figure 4: Overview of our three proposed methods: (a) WSP, (b) BWSP, and (C) ABWSP.

and column-wise WSP (WSP-C), which remove row and column vectors respectively. This methods remove the row vector ($\mathbb{R}^{1 \times c}$) and column vector ($\mathbb{R}^{n \times 1}$) of the sub-GEMM, respectively. In the Section 5.1, we performed an analysis on two techniques (Row type and Column type) and found that Row type is more effective.

$$S = A^T [StructuredPrune(GgG^T) \odot (B^T dB)] A \qquad (4)$$

## 4.2 BALANCED WINOGRAD STRUCTURED PRUNING (BWSP)

**(a) Motivation**  WSP exhibits an imbalanced characteristic attributed to the different pruning ratios ($PR$) in each of the 16 ($p^2$) sub-GEMMs. This imbalance presents a significant bottleneck for parallel processors, such as GPUs, as each sub-GEMM is assigned to a SM for independent execution (Li et al., 2019). In particular, SMs handling sub-GEMMs with larger $PR$s tend to complete their computations before SMs assigned to sub-GEMMs with smaller

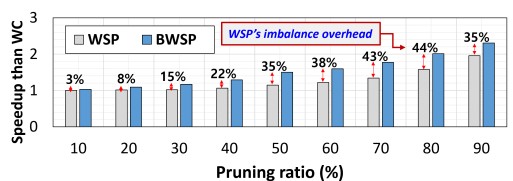

Figure 5: Comparison of speedup (inference time) of WSP and BWSP on ResNet-18.

$PR$s, leading to idle periods (see Figure 4(a)). Such an idle period delays the execution of the following layers. In Figure 5, the BWSP has an average speed improvement of 27% on ResNet-18 compared to WSP.

**(b) Method**  In order to address the issue of imbalance, we have enhanced WSP by introducing a new approach called Balanced WSP (BWSP) (see Figure 4(b)). With BWSP, we ensure that all sub-GEMMs have an equal $PR$. To achieve this equal $PR$ across all sub-GEMMs, we employ a specific order for *pruning*. When performing the *pruning* process, we utilize a similar approach to the one used in WSP, which takes into account small representative values calculated using the L2 norm. However, in BWSP, these representative values are considered within each sub-GEMM individually, rather than being assessed across all GEMMs collectively.

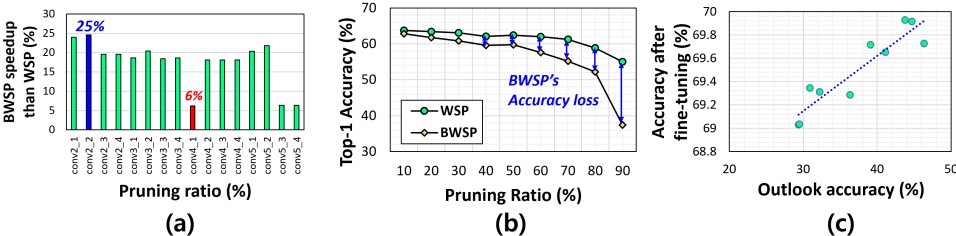

Figure 6: ResNet-18 on ImageNet. (a) Comparison of performance improvements in layers obtained from applying balancing on WSP. (b) Comparison of Top-1 accuracy between WSP and BWSP. The fine-tuning epoch is one. (c) Correlation of outlook accuracy and fine-tuning accuracy.

### 4.3 ADAPTIVE BALANCED WINOGRAD STRUCTURED PRUNING (ABWSP)

**(a) Motivation** BWSP enhances the load balancing in Batched GEMM operations by emphasizing regularity in the sparse weights of the WSP concept. Figure 6(a) illustrates the advantages of BWSP, up to 25% better. A distinctive acceleration is achieved across entire layers by ensuring an evenly distributed sparsity in the weight matrices involved in Batched GEMM. However, Figure 6(b) reveals a trade-off: The accuracy of BWSP exhibits a disadvantage compared to relying on WSP alone. This observation is understandable since significant values need to be unintentionally excluded in network pruning. To maximize accelerating inference and preserving accuracy, we introduce an adaptive framework that applies BWSP or WSP selectively, considering evaluation accuracy and speed-up due to workload balance, simultaneously.

**(b) Method** We introduce the Adaptive Balanced Winograd Structured Pruning (ABWSP) method (see Figure 4(c)). ABWSP is an auto algorithm that automatically adjusts the number of layers applied to BWSP or WSP when the accuracy loss of the BWSP pruned model becomes significant. ABWSP operates in two main ways: Pre-processing and Evaluation & Pruning.

**Pre-processing** Looking at Figure 4 (c), first, we measure the speedup of all the layers from BWSP (called Balanced Speedup (BS)). Second, we search the pruning candidates that may result from applying either WSP or BWSP to each layer. If the number of pruning candidates is too large, we use Monte Carlo sampling to reduce the cases.

**Evaluation & Pruning** Initially, the layer-by-layer BS obtained from pre-processing is used to calculate the total BS for each case. Next, an adaptive-BN-based evaluation method (Li et al., 2020) is applied to ABWSP to measure the "outlook accuracy" of each candidate. This evaluation method uses minimal training iteration and validation data (e.g., 100-iteration). As seen in Figure 6(c), outlook accuracy and accuracy are highly correlated. Finally, the one with the highest BS and within the top $\alpha\%$ of outlook accuracy is chosen as the winner and undergoes fine-tuning. After fine-tuning, if the accuracy drop of the winner is negligible, the winner is added to the pruned model list. After about $\beta$-iterations of ABWSP auto algorithm, the best model from the pruned model list is selected as the final pruned model.

## 5 EXPERIMENTS

**Baseline settings** We evaluate ABWSP using NVIDIA RTX 3090 GPUs. We use the pre-trained CNN models from Pytorch framework (Paszke et al., 2019). We perform fine-tuning with only $40$ epochs. We use SGD optimizer with the weight decay, $1 \times 10^{-4}$ and the momentum as $0.9$, for fine-tuning. We set a batch size of $256$. The initial learning rate is set to $0.11$ and divide it by $10$ every 30 epochs. To accelerate *WC*, we exploit the GPU kernel based on the API (Guo et al., 2020).

**Model and dataset** We evaluate the performance of ABWSP using ImageNet dataset (Deng et al., 2009) and CIFAR-10 (Krizhevsky et al., 2009). CIFAR-10 contains $50K$ training images and $10K$ test images for 10 classes. ImageNet contains $1.28M$ training images and $50K$ validation images for 1000 object classes. For the validation of image classification, we assess our method with CNN models: VGG-16 (Simonyan & Zisserman, 2015) and ResNet-18, 20, 32, and 56 (He et al., 2016).

| Method | $PR$ (%) | Top-1 (%) | Speedup | Method | $PR$ (%) | Top-1 (%) | Speedup |
|--------|----------|-----------|---------|--------|----------|-----------|---------|
| **BWSP** | 50 | 69.9 | 1.12× | **BWSP** | 50 | 69.8 | 1.49× |
| **Column** | 60 | 69.7 | 1.13× | **Row** | 60 | 69.6 | 1.60× |
| **[ours]** | 70 | 69.7 | 1.15× | **[ours]** | 70 | 69.4 | 1.77× |

Table 1: Comparison of accuracy and speedup (inference time) than *WC* of ResNet-18 on ImageNet.

## 5.1 BWSP-Column vs. BWSP-Row

Our proposed methods are classified into two distinct categories based on the vector direction: Column type (BWSP-C) and Row type (BWSP-R). Two analysis experiments are conducted for comparison of BWSP-C and BWSP-R. First, in Table 1, we compare accuracy according to the pruning ratio ($PR$) of BWSP-R and BWSP-C. Since the pruning unit size of BWSP-R and BWSP-C is almost similar, accuracy according to the $PR$ is also similar. Second, both techniques show speed improvement as $PR$ increases. However, the speedup achieved by BWSP-R is greater than that of BWSP-C. Therefore, we use the BWSP-R when comparing with other models.

## 5.2 ABWSP vs. BWSP vs. WSP

In Figure 7(a), we present a comprehensive comparison of the three proposed models - WSP, BWSP, and ABWSP, considering multiple metrics such as $PR$, accuracy, and speed. As depicted in Figure 7(a), at a $PR$ of 80%, WSP and ABWSP have an accuracy drop of less than -1%, while BWSP experiences the most significant accuracy drop at -1.27%. Interestingly, at the same 80% $PR$, ABWSP outperforms WSP in terms of speed, being approximately 1.20 times faster. As a result of this comprehensive comparison, ABWSP is the most effective approach for ResNet18, when compared with the other our methods.

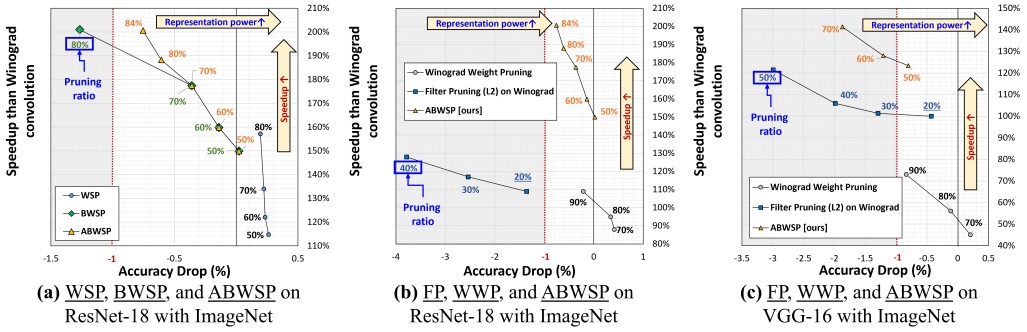

**(a)** WSP, BWSP, and ABWSP on ResNet-18 with ImageNet

**(b)** FP, WWP, and ABWSP on ResNet-18 with ImageNet

**(c)** FP, WWP, and ABWSP on VGG-16 with ImageNet

Figure 7: Comparison of speedup (inference time) and accuracy in various pruning ratio. We set $\alpha$ and $\beta$ of ABWSP to 30 and 5, respectively.

## 5.3 Comparison with Baselines: FP and WWP

We compare our proposed ABWSP with previously proposed pruning methods (i.e., WWP and FP) in terms of accuracy, $PR$, and speedup.

**Comparison with Filter Pruning (FP)** As shown in Figure 7(b) and (c), FP and ABWSP are faster than a dense model because FP and ABWSP keep a structured form after FTrans. However, the pruning unit size of FP is too large to avoid accuracy drop because FP creates a structured form without considering the *WC*. Since ABWSP is a pruning method considering *WC*, the pruning unit size of ABWSP is about $p^2$ times less than that of FP. In ResNet-18, the FP exceeds -2.0% accuracy drop from 30% $PR$ (see Figure 7(b)). On the other hand, even if ABWSP prunes up to 80% $PR$, the accuracy drop of ABWSP does not exceed -0.6% in ResNet-18. In VGG-16 and ResNet-18, within -2.0% accuracy drop, ABWSP is 1.34× and 1.84× faster than FP, respectively (see Figure 7(a) and (b)).

**Comparison with Winograd Weight Pruning (WWP)** Both WWP and ABWSP are pruning techniques considering *WC*. Since the pruning unit size of ABWSP is about $c$ times larger than that of WWP, WWP can prune more precisely than ABWSP. As shown in Figure 7(c), when the $PR$ is 70%

| Model | Method | O-Err. | P-Err. | Gap | $PR$ | Speedup |
|---|---|---|---|---|---|---|
| ResNet-20 | SCOP (Tang et al., 2020) | 7.78% | 9.25% | -1.47% | 56% | 1.47× |
| | GCNP (Di Jiang & Yang, 2022) | 7.75% | 8.42% | -0.67% | 39% | 1.18× |
| | **WSP [Ours]** | 8.27% | 8.88% | -0.61% | 85% | **1.57×** |
| ResNet-32 | SFP (He et al., 2018) | 7.37% | 7.92% | -0.55% | 42% | 1.19× |
| | FPGM (He et al., 2019) | 7.37% | 8.07% | -0.70% | 53% | 1.44× |
| | SCOP (Tang et al., 2020) | 7.34% | 7.87% | -0.53% | 56% | 1.49× |
| | **WSP [Ours]** | 7.37% | 7.74% | -0.37% | 83% | **1.61×** |
| ResNet-56 | Hrank (Lin et al., 2020) | 6.74% | 6.83% | -0.09% | 42% | 1.21× |
| | SCOP (Tang et al., 2020) | 6.30% | 6.36% | -0.06% | 56% | 1.50× |
| | GCNP (Di Jiang & Yang, 2022) | 6.28% | 7.25% | -0.97% | 71% | 1.60× |
| | **WSP [Ours]** | 6.99% | 7.69% | -0.70% | 90% | **2.04×** |

Table 2: Comparison of speedup (inference time) on ResNet-20, 32, and 56 with CIFAR-10. O-Err. denotes errors of the pre-trained model. P-Err. denotes errors of the pruned model. Gap is the difference in size between O-Err. and P-Err.. $PR$ denotes pruning ratio.

| Model | Method | Top-1 ↓(%) | $PR$ (%) | Inference time ($ms$) | Index ratio | Speedup than $WC$↑ |
|---|---|---|---|---|---|---|
| ResNet-18 on ImageNet | *WC* (Lavin & Gray, 2016) | - | - | 3.73 | - | 1.00× |
| | TAS (Dong & Yang, 2019) | -1.50 | 33.3 | 3.21 | - | 1.16× |
| | LCCL (Dong et al., 2017) | -3.65 | 34.6 | 3.19 | - | 1.17× |
| | **ABWSP [Ours]** | **0.19** | **40.1** | 3.09 | 5% | **1.20×** |
| | SFP (He et al., 2018) | -3.18 | 41.7 | 3.06 | - | 1.22× |
| | FPGM (He et al., 2019) | -1.87 | 41.7 | 3.06 | - | 1.22× |
| | PFP (Liebenwein et al., 2020) | -2.36 | 43.8 | 3.03 | - | 1.23× |
| | **ABWSP [Ours]** | **-0.14** | **60.7** | 2.33 | 7.6% | **1.60×** |
| | **ABWSP [Ours]** | **-0.36** | **70.3** | 2.11 | 7.6% | **1.77×** |
| | **ABWSP [Ours]** | **-0.75** | **84.0** | 1.86 | 7.7% | **2.01×** |

Table 3: Comparison of inference time ($ms$) with FP on *WC*. Top-1↓ denotes accuracy drop. Index denotes index computation kernel time. $PR$ denotes pruning ratio.

in VGG-16, WWP has 2.0% higher accuracy than ABWSP, respectively. However, despite pruning up to 90% $PR$ in VGG-16, WWP is slower than the *WC* dense model due to the significant index computation of WWP. Unlike WWP, ABWSP maintains a structural form, so ABWSP is 2.01× faster than *WC* when the $PR$ is 84% (see Figure 7(b)).

## 5.4 COMPARISON WITH STATE-OF-THE-ART FP

**CIFAR-10** In Table 2, we evaluate our method using ResNet (20, 32, and 56) on the CIFAR-10. In ResNet-20, when the $PR$ of SCOP (Tang et al., 2020) is 56%, the accuracy drop is more than 1%. On the other hand, WSP's accuracy drop is -0.61% in 85% $PR$. In ResNet-32, WSP is the highest $PR$ at 83%, and the accuracy drop is the smallest than other FP. WSP at 90% $PR$ has 1.28× faster and 0.27% higher accuracy than GCNP (Di Jiang & Yang, 2022) at 71% $PR$, in ResNet-56.

**ImageNet** In Table 3, we evaluate our method using ResNet-18 on the ImageNet. When the $PR$ of LCCL (Dong et al., 2017) and SFP (He et al., 2018) is 34.6% and 41.7%, respectively, the accuracy drop is more than 3%. On the other hand, ABWSP's accuracy drop is only less than 2% in 70% $PR$. Due to the index computation of ABWSP, FPGM (He et al., 2018) is about 1.01× faster than ABWSP when ABWSP and FPGM are at almost same 40% $PR$. ABWSP at 70% $PR$ has 1.33× faster and 0.79% higher accuracy than PFP (Liebenwein et al., 2020) at 43.8% $PR$.

## 6 CONCLUSION

We propose ABWSP that accelerates *WC* in consideration of *WC* computational characteristics on a parallel processor. We perform vector-wise pruning in BGEMM operation, which brings delicate structured data pattern compared to WWP and FP, for accuracy maintenance and speedup. ABWSP also makes balanced matrices for computing utilization. ABWSP has realistic CNN representation power at a high $PR$ by selectively adapting the best algorithm in each layer.

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

## A  APPENDIX: LIMITATION OF MIDDLE-GRAINED PRUNING ON WC

The Winograd transformation process converts sparse matrices into dense matrices thus it is challenging to apply Winograd convolution (*WC*) and *pruning* techniques, simultaneously (Liu et al., 2018). This is because the Filter Transformation (FTrans) process densely changes the sparsity of pruned model by *pruning*. FTrans is performed in units of 2D tensor ($\mathbb{R}^{k_h \times k_w}$) out of 4D tensor ($\mathbb{R}^{n \times c \times k_h \times k_w}$) of the convolution layer. Therefore, *pruning* with a pruning unit size of 2D tensor ($\mathbb{R}^{k_h \times k_w}$) or larger can be compatible simultaneously with *WC*. Filter pruning (FP), which is pruned in units of 3D tensor ($\mathbb{R}^{c \times k_h \times k_w}$), can be compatible with *WC* (Yu et al., 2019). Middle-grained pruning (e.g., $1 \times n$ Pruning (Lin et al., 2022) and Block Sparse (Narang et al., 2017)) with a pruning unit size of more than 2D tensor ($\mathbb{R}^{k_h \times k_w}$) can also be compatible with *WC*. As shown in Figure 8, the middle-grained pruning prunes in units of vector or block when the convolution layer is converted to matrix multiplication using im2col (called lowering method) (Chellapilla et al., 2006; Chetlur et al., 2014). When the middle-level pruned model is converted to col2im, most of them are pruned in 2D tensor units ($\mathbb{R}^{k_h \times k_w}$). Therefore, middle-grained pruning also be compatible with *WC*, like FP. However, unlike FP, the middle-grained pruned model have unstructured data pattern in Winograd-domain. As a result, middle-grained pruning is still difficult to improve performance like FP or our proposed ABWSP without an appropriate GPU library.

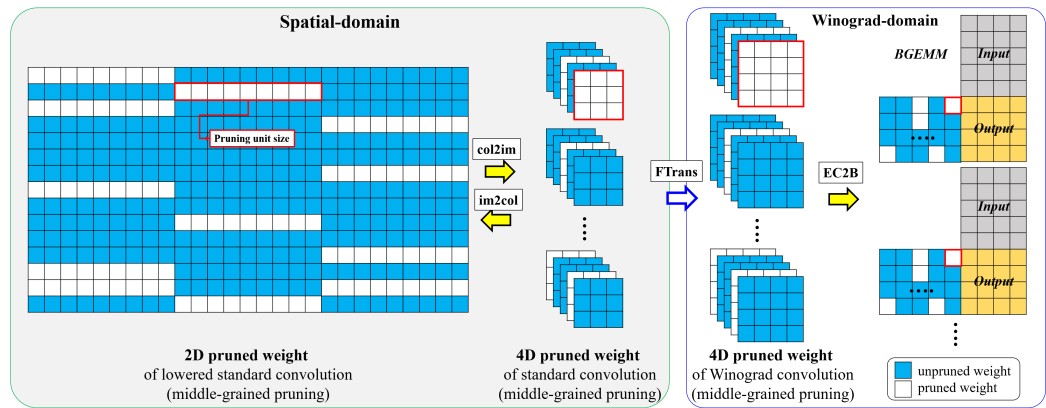

Figure 8: Overview of middle-grained pruning on *WC* and standard convolution

# B   APPENDIX: LARGE PRUNING UNIT SIZE PROBLEM OF FP

## B.1   PRUNING UNIT SIZE ON VGG-16

When the size of the pruning unit increases, there is a notable decrease in the representation power of the network. This is due to the fact that the pruning unit size plays a crucial role in determining the precision of the *pruning* process. As shown in Figure 9, in the case of VGG-16, FP ($p^2 \times \mathbb{R}^{1 \times c}$) has a larger pruning unit size than WSP-R ($\mathbb{R}^{1 \times c}$) in all layers except for the first convolution layer. For example, on VGG-16, the pruning unit sizes of FP on Winograd-domain are either $1,024$, $2,048$, $4,096$, and $8,192$ parameters depending on the layers, except the first convolution layer. On the other hand, the pruning unit size of WSP-R is almost 16 ($p^2$) times smaller than FP's, because the pruning unit size of WSP-R are either $64$, $128$, $256$, and $512$ parameters depending on the layers. Therefore, our proposed WSP and ABWSP in VGG-16 are more sophisticated *pruning* approaches than FP in terms of accuracy and inference speed.

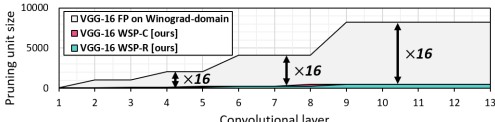

Figure 9: Comparison of pruning unit size of pruned model on VGG-16.

## B.2   APPENDIX: COMPARISON OF ACCURACY AND PRUNING RATIO

**ResNet-18 on ImageNet, Fine-tuning epoch is 40**   The side-effect of FP has representation loss when the models are aggressively pruned. As shown in Figure 10, the pruning unit size of FP on the Winograd-domain is $p^2 \times \mathbb{R}^{1 \times c}$, which tends to be too large, so it removes the important weights with redundant weights together. In Figure 10, when evaluating FPs with $PR$s of $30\%$ in ResNet-18, the Top-1 accuracy is reduced by $3\%$. In 40 fine-tuning epoch, both WWP and WSP have an Top-1 accuracy drop of less than $3\%$ in $90\%$ $PR$. In particular, FP has an Top-1 accuracy drop of more than $3\%$ in most $PR$s. Even though FP keeps the matrices in a structured form to leverage the full computing capabilities of existing general-purpose computing devices, FP shows significant accuracy drops in the high $PR$ even with retaining.

# C   APPENDIX: EFFECT OF FINE-TUNING

We evaluate the accuracy according to the variation of fine-tuning epoch with four different *pruning* on ResNet-20 and VGG-16 in CIFAR-10 (see Figure 11 and 12).

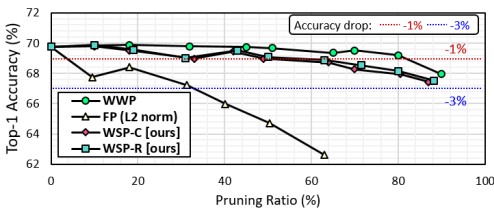

Figure 10: Comparison of accuracy and $PR$. Top-1 validation and $PR$ for four model on a variation of ResNet-18 on ImageNet. The fine-tuning epoch is 40.

**ResNet-20 on CIFAR-10**  Since WWP is fine-grained pruning, when epoch is 10 and 100, only a minimum accuracy drop (less than $0.64\%$ and $0.03\%$) is observed even with $70\%$ of pruning ratio ($PR$). In Winograd-domain, FP is a method of removing more than hundreds of elements from ResNet-20, so even if the $PR$ is $10\%$ when fine-tuning epoch is 1 and 10, significant accuracy degradation (at least $1.65\%$) is observed. Even if the FP pruned model is fine-tuned more than 100 epochs, if the $PR$ exceeds $40\%$, the accuracy drop exceeds $1.72\%$. Since the pruning unit size of WSP is $p^2$ times less than FP's, WSP can prune more sophisticated than FP. With only 10 fine-tuning epochs, our proposed WSP shows an accuracy drop of less than $1.5\%$ even with a higher than $60\%$ of $PR$.

**VGG-16 on CIFAR-10**  Since WWP is fine-grained pruning, when epoch is 10, only a minimum accuracy drop (less than $0.6\%$) is observed even with $70\%$ $PR$. On the other hand, FP is a method of removing thousands of elements from VGG-16, so even if the $PR$ is $20\%$ when fine-tuning epoch is 1 and 10, Significant accuracy degradation (at least $5\%$) is observed. Even if the FP pruned model is fine-tuned more than 100 epochs, if the $PR$ exceeds $30\%$, the accuracy drop exceeds $5\%$. Since the pruning unit size of WSP is $\mathbb{R}^{n \times 1}$ or $\mathbb{R}^{1 \times c}$, WSP can prune more sophisticated than FP.

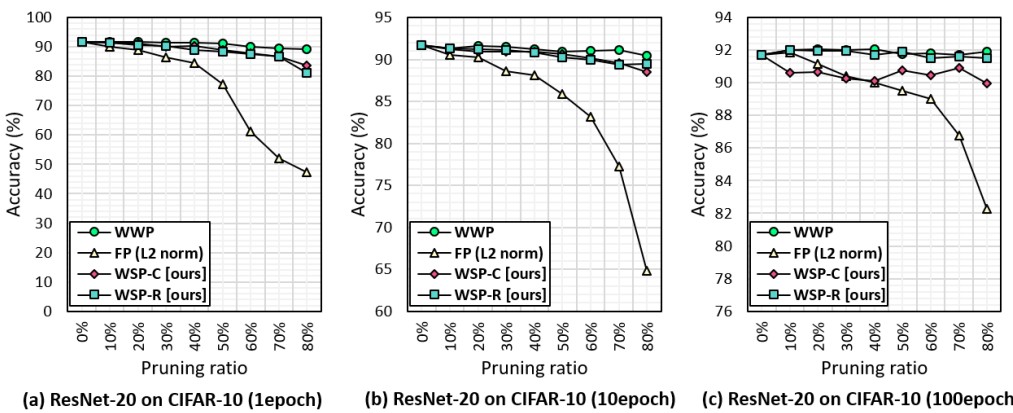

**(a) ResNet-20 on CIFAR-10 (1epoch)**  **(b) ResNet-20 on CIFAR-10 (10epoch)**  **(c) ResNet-20 on CIFAR-10 (100epoch)**

Figure 11: Comparing WSP with WWP and FP of ResNet-20 on CIFAR-10. We experiment with $PR$ and accuracy using three epochs of fine-tuning: 1, 10, and 100.

# D  APPENDIX: PRUNED FILTER VISUALIZATION

We use a visualization method to understand that WSP has a regular data pattern and middle pruning unit size at *WC*. In Figure 13, we sequentially visualized WWP (Liu et al., 2018), WSP-C, and WSP-R. The pruned filter visualization is done with the $PR$ of $50\%$ within the $1\%$ variance. For visualization, we convert the EWMM 4D weight matrix of the WC to BGEMM weight matrix using EC2B converting method. The weight map of WWP shows irregular data patterns in both non-converted and converted weight matrices as shown in Figure 13(a) and 13(d). WSP-C has the same pruning mask between matrices with the same output channel as shown in Figure 13(b). In

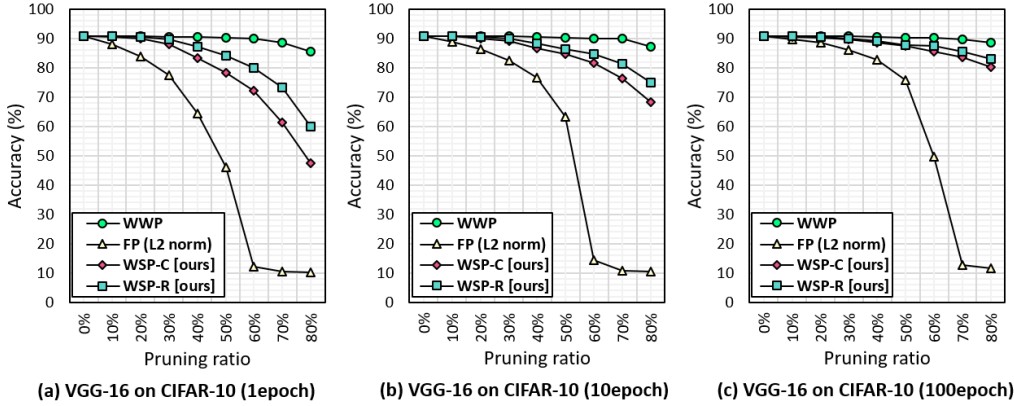

Figure 12: Comparing WSP with WWP and FP of VGG-16 on CIFAR-10. We experiment with $PR$ and accuracy using three epochs of fine-tuning: 1, 10, and 100.

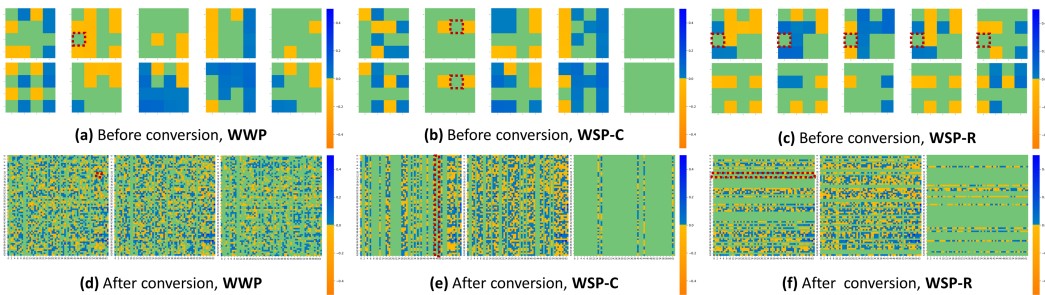

Figure 13: Pruned filter visualizations in res2a_2a layer of ResNet-18 on ImageNet. Positive, negative and pruned weights are in blue, yellow and green respectively. Dotted redline denotes a pruning unit.

Figure 13(e), the WSP-C shows regular data patterns which have column-wise vector pruning unit at converted weight. In Figure 13(c), WSP-R has the same pruning mask between matrices with the same input channel. WSP-R has row-wise vector pruned data pattern as shown in Figure 13(f).

