# OpenReview forum: "Winograd Structured Pruning"
_ICLR.cc/2024/Conference — ICLR 2024 Conference Withdrawn Submission_

### Official Review · Reviewer_wzFL · 2023-10-31

**Soundness:** 3 good
**Presentation:** 2 fair
**Contribution:** 2 fair
**Rating:** 3
**Confidence:** 4

**Summary:**

This paper proposes an optimization for efficient Winograd structure pruning to achieve higher parallelism and workload balance while maintaining good accuracy. It consists of two major optimizations: efficiently utilizing the computing units on GPUs by simultaneously pruning grouped weights and maintaining an equal pruning ratio between WC matrices to ensure that each GPU computing unit completes the same number of workloads. Compared to filter pruning and the vanilla Winograd algorithm, it achieves better accuracy or lower latency.

**Strengths:**

- Achieves notable performance gain (higher accuracy or lower latency) compared with the baselines
- Evaluated two representative datasets and two different network structures.

**Weaknesses:**

- Excessive use of abbreviations can make certain chapters challenging to comprehend. For instance, "FP" is introduced in the introduction without any explanation until the related work section.
- Figure 1 is a little bit complicated, making it difficult to grasp the underlying core ideas.
- The main concerns revolve around novelty. The pruning technique and proposed workload balancing optimization are incremental contributions. Recently, there have been numerous works on pruning that achieve impressive compression ratios [1], but it remains unclear how much benefit they surpass unstructured pruning (with a higher pruning rate). The third contribution is also straight-forward.

[1] AutoCompress: An Automatic DNN Structured Pruning Framework for Ultra-High Compression Rates

**Questions:**

Please see detailed comments as above

---

### Official Review · Reviewer_wyHk · 2023-10-31

**Soundness:** 3 good
**Presentation:** 3 good
**Contribution:** 3 good
**Rating:** 6
**Confidence:** 4

**Summary:**

This paper presents Adaptive Balanced Winograd Weight Pruning (ABWSP), a method that prunes weights in Winograd Convolution on GPUs. ABWSP considers three key factors: properly selecting the optimal pruning unit size that is GPU-aware, maintaining workload balance between matrices by a uniform pruning ratio, and toggling between two pruning policies (BWSP and WSP) based on layer importance. Evaluation results show that the proposed ABWSP outperforms two baselines, FP (for latency and accuracy) and WWP (for latency).

**Strengths:**

+ This work targets an important kernel in CNNs.
+ The GPU-aware designs and optimizations are meaningful for this research direction.
+ The Winograd Convolution is carefully introduced.
+ The figures are carefully drawn and very helpful.
+ The proposed ABWSP is evaluated on ImageNet and compared with two baselines.

**Weaknesses:**

- Novelty is somewhat incremental. Structured pruning is somewhat well-known.
- The relationship between this work and WWP+varied sparse tensor optimizations on GPUs is somewhat unclear. It seems a direct comparison is missing.
- A proper ablation study on individual Conv layers would be helpful.
- It contains some minor presentation issues.

**Questions:**

Overall, this is a carefully written paper. The proposed structured pruning of Winograd generally makes sense. It would be extremely helpful if the authors could justify their contributions from these aspects:

1. About its novelty: structured (and semi-structured) model pruning is a research hotspot with extensive research efforts and results. It would be helpful to further justify the difference (and novelty) of performing structured pruning on Winograd Conv from other operators.

2. Fig. 7 shows that ABWSP cannot achieve the same accuracy as WWP with an identical pruning ratio. In other words, it sacrifices accuracy for latency performance gains. This is very similar to the accuracy and latency tradeoff on other operators. There exist many optimization techniques on GPUs for irregular/sparse tensor computations. It is somewhat unclear if the WWP implementation is fully optimized, particularly without the necessary implementation details in the current version of the paper. It would be extremely helpful to offer more details about its implementation, such as data storage format, any applied optimizations, etc.

3. It would be helpful to provide a speedup ablation study on an individual Conv layer with varied configurations for performance understanding.

Minor:
Please consider revising these minor issues:
- Some important abbreviations (e.g., WSP, BWSP, FP) are used without explanations when they are introduced.
- Font sizes in some important figures are too small (e.g., Fig. 7) to read.

---

### Official Review · Reviewer_oE96 · 2023-11-05

**Soundness:** 3 good
**Presentation:** 1 poor
**Contribution:** 3 good
**Rating:** 6
**Confidence:** 3

**Summary:**

This study proposes Adaptive Balanced Winograd Structured Pruning (ABWSP), a method for weight pruning in Winograd convolution networks on GPU devices. The ABWSP algorithm optimizes the pruning process by considering three factors, including the size of the pruning unit, workload balance, and the importance of different model layers. The effectiveness of ABWSP and its advantages over existing benchmark methods are demonstrated through extensive experiments on multiple datasets.

**Strengths:**

The proposed ABWSP is a novel pruning method that balances accuracy and speed-up by adaptive applying BWSP or WSP to different layers. The overall quality of the manuscript is good and the English usage is satisfactory. The motivation of this study is intuitive and reasonable, which may be inspiring to researchers in this specific field.

**Weaknesses:**

The major weakness of this manuscript in its current status is the organization of content and clarity of writing. There are many abbreviations used, which makes the manuscript unfriendly to readers who are not experts in this specific field. Also, the authors may want to proofread their manuscript and improve the clarity and conciseness, and try not to distract the readers' attention by introducing technical details.

**Questions:**

Please refer to the weakness section.

---

### Official Review · Reviewer_iq1J · 2023-11-10

**Soundness:** 2 fair
**Presentation:** 3 good
**Contribution:** 2 fair
**Rating:** 3
**Confidence:** 4

**Summary:**

This paper proposes Adaptive Balanced Winograd Structured Pruning (ABWSP) to prune weights in WC networks executed on GPUs. The evaluation shows ABWSP outperforms FP and WWP on ImageNet.

**Strengths:**

1. The figures are drawn well.
2. It is good to evaluate large datasets, like ImageNet.

**Weaknesses:**

1. The discussion of related works is confusing. The authors mention that FP has no index computation overhead and leads to regular patterns, while all the WWP methods create an irregular data pattern and require significant index computation. From this argument, we can see FP outperforms WWP with hardware-efficient sparsity and no index computation. So why do we not use FP directly rather than proposing ABWSP to apply a hardware-efficient sparsity scheme in WC to reduce the overhead of index computation?
3. Evaluation is not sufficient. Most of the evaluation focuses on the ResNet and VGG architectures. It will be better if more model architectures are evaluated, like EfficientNet, DenseNet, MobileNet, and ShuffleNet.
4. Baselines are restricted. Only WWP and FP are considered as baselines, missing a lot of SOTA pruning works [1-9]. Also, the related WWP methods discussed in Section 3 should also be compared in the Evaluation.
5. The ablation study is missing.

[1] EigenDamage: Structured Pruning in the Kronecker-Factored Eigenbasis

[2] Layer-adaptive sparsity for the magnitude-based pruning

[3] Automatic Attention Pruning: Improving and Automating Model Pruning using Attentions

[4] Chipnet: Budget-aware pruning with heaviside continuous approximations

[5] Provable filter pruning for efficient neural networks

[6] Accelerate CNNs from Three Dimensions: A Comprehensive Pruning Framework

[7] EagleEye: Fast Sub-net Evaluation for Efficient Neural Network Pruning BT

[8] DMCP: Differentiable Markov Channel Pruning for Neural Networks

[9] GDP: Stabilized Neural Network Pruning via Gates with Differentiable Polarization

**Questions:**

1. In the Introduction, the authors claim that pruning is developed without considering WC, resulting in the removal of sparsity in fine-grained pruning. However, if such pruning doesn't result in performance degradation, as demonstrated by state-of-the-art (SOTA) pruning research (above [1-9]), then the loss of sparsity can be acceptable, is it right?
2. Can the authors compare with more SOTA pruning works and provide the ablation study?